# Survival of dental implants in irradiated head and neck cancer patients compared to non-irradiated patients: An umbrella review

Maria Cândida Dourado Pacheco[1], Alexandre Henrique dos Reis Prado[2], Danilo Viegas da Costa[3], Lara Cancella de Arantes[2], Caio Fernando Teixeira Portela[1]*, Tarcísio Passos Ribeiro Campos[1], Arno Heeren de Oliveira[1]

1 Department of Nuclear Engineering, Federal University of Minas Gerais, School of Engineering, Belo Horizonte, Minas Gerais, Brazil, 2 Restorative Dentistry, Federal University of Minas Gerais, School of Dentistry, Belo Horizonte, Minas Gerais, Brazil, 3 Dental Clinics with an Emphasis on Periodontics at the Pontifical Catholic University of Minas Gerais, Belo Horizonte, Minas Gerais, Brazil

☯ These authors contributed equally to this work.

* caiofernando\protect_fisica@yahoo.com.br

**Data availability statement:** All relevant data are within the paper and its Supporting information files.

## Abstract

**Background:** There is an increasing demand for oral rehabilitation in patients undergoing irradiation for head and neck cancer treatment. Although radiotherapy appears to adversely affect implanted oral rehabilitation, the evidence provided by previous systematic reviews in the field remains controversial. Thus, this umbrella review aimed to evaluate the survival of dental implants installed before and/or after radiotherapy in patients with head and neck cancer compared to those not irradiated.

**Methods:** A comprehensive search of electronic databases was performed in the PubMed, Cochrane Library, Embase, Web of Science, and Google Scholar databases, including manual searches, from inception to February, 2024, with no language or date restrictions. The Preferred Reporting Items for Systematic Reviews and Meta-Analyses (PRISMA) 2020 guidelines were followed. Survival percentage of dental implants was the primary outcome. A Measurement Tool to Assess systematic Reviews (AMSTAR) 2 was used as a critical appraisal tool for the included studies. The protocol for this review has been registered with PROSPERO (CRD42023406059).

**Results:** Of the 1,811 articles screened, 11 articles that evaluated 73,674 implants were included. Quantitative analyses showed 2,674 failures out of 14,471 implants installed in irradiated bone and 1,825 failures out of 34,092 implants in non-irradiated bone. The survival rate was 81.52% for irradiated implants and 94.64% for non-irradiated implants. A significant difference in survival in favor of implants in non-irradiated bone is supported by 11 meta-analyses. The included systematic reviews showed critically low methodological quality.

**Funding:** The author(s) received no specific funding for this work.

**Competing interests:** The authors have declared that no competing interests exist.

**Conclusions:** Although the included studies had low methodological quality, the findings indicate a higher failure rate of implants in irradiated patients. Further well-designed clinical studies are warranted.

## 1 Introduction

Osseointegrated dental implants are widely used and can provide a safe treatment option for edentulous patients, demonstrating promising survival rates within a 5 to 10-year follow-up period [1–3]. However, the risks associated with implant failures have been extensively investigated over the years [4–11]. Possible reasons for failure are low insertion torque of implants planned to be loaded immediately or early, inexperienced surgeons inserting the implants, insertion of the implant into the maxilla, the patient history (radiotherapy, bruxism, periodontitis and early implant loss) [12,13].

The incidence of head and neck cancer has been increasing among people worldwide, according to data presented by the Global Cancer Observatory (GLOBOCAN), incidence of head neck cancer for the year 2022 was approximately around 1.5 million new cases with around 500,000 deaths from the disease [14]. There is an increasing demand for oral rehabilitation in patients undergoing irradiation for head and neck cancer treatment [15], which is also associated with the increased frequency of dental implants installed in these patients [16]. However, radiotherapy seems to adversely affect implanted oral rehabilitation [17–19] due to radiation⊠induced fibroatrophic process [20], leading to tissue hypoxia, hypocellularity, hypovascularity and reduced bone regeneration capacity [21].

Previous umbrella reviews [22,23] have recognized oral rehabilitation with implants as a feasible treatment option for patients who have undergone radiotherapy. However, despite numerous systematic reviews investigating the survival of dental implants in head and neck cancer patients receiving radiotherapy - incorporating various clinical analyses - the findings remain inconclusive [24–26].

Futhermore, due to the heterogeneity of studies and the absence of comparator groups in most included reports, it is recommended that the presented results be interpreted with caution. To address inconsistencies in study methodologies and ensure objectivity in research, Cochrane introduced the umbrella review as a study design that consolidates evidence from multiple systematic reviews into a single comprehensive analysis [27,28].

This approach facilitates more reliable decision-making by systematically comparing scientific data. Therefore, in light of the conflicting results, the present umbrella review aimed to analyze the available relevant systematic reviews, comparing irradiated versus non-irradiated patients, in an attempt to determine whether there is an influence of head and neck radiotherapy on dental implant survival in patients with head and neck cancer.

## 2 Materials and methods

This study followed the guidelines of the Preferred Reporting Items for Systematic Reviews and Meta-Analyses (PRISMA) 2020 (Page et al., 2021) [21]. The research protocol was registered on the International Prospective Register of Systematic Reviews (PROSPERO) database under the number CRD42023406059.

### 2.1 PICO formulation

The PICO (Population; Intervention; Comparison; Outcome) [29] strategy was used to formulate the following research question: "What is the survival of dental implants installed

before and/or after radiotherapy in patients (male and female) with head and neck cancer, compared to those not exposed to radiation?" PICO information can be seen in Table 1, in the S1 Text (S1 Table).

## 2.2 Eligibility criteria

The inclusion criterion were systematic reviews, with or without meta-analysis, that assessed the survival of dental implants installed before and/or after radiotherapy in patients with head and neck cancer, compared to patients with head and neck cancer in whom implants were installed in non-irradiated bone. However, systematic reviews of animal studies, other types of reviews different from systematic reviews, and studies for which the full text was unavailable were excluded.

## 2.3 Research sources and search strategy

The electronic search was conducted by two independent reviewers (MCDP and LCA), including studies published in PubMed, Cochrane Library, Embase, and Web of Science, from their inception until February 23, 2024. Additionally, grey literature search was searched in Google Scholar database [30]. Search strategies were developed for each database using indexed terms and synonyms (S1 Text, S1 Table). A manual search of references was performed in all included studies. In case of doubt, a third reviewer (DVC) was consulted. Team members are experts in study translation, so there were no restrictions on the language or year of publication for the articles.

## 2.4 Study selection process

The selection of studies was conducted by two reviewers (MCDP and LCA) independently, in a two-stage process using the Rayyan QCRI website (Rayyan, Qatar Computing Research Institute, Al-Rayyan, Qatar) [31,32]. Duplicate studies were excluded using reference management software (EndNote® X9; Clarivate, Philadelphia, PA, USA). In the first stage, titles and abstracts were assessed, and in the next stage, the full text selected from the titles/abstracts was read, and those that did not meet the eligibility criteria were excluded. Disagreements were resolved by a third reviewer (DVC). Contacting authors to clarify doubts was not necessary. The level of agreement between the reviewers was assessed using Cohen's Kappa (k= 0,90) [33], indicating an almost perfect level of agreement.

## 2.5 Data extraction and analysis

Data were collected by two reviewers (MCDP and LCA) independently in April 2024, based on the main characteristics of the studies, in an Excel form created and customized for the current review and included: study and publication year, number of participants/age range, country of publication, number of included studies/study design, presence of meta-analysis, variation in radiation dose (Gy)/implant installation time, number of failed/installed

**Table 1. PICO formulation.**

| P | Population | Patients undergoing radiotherapy for head and neck cancer |
|---|---|---|
| I | Intervention | Installation of dental implants before and/or after radiotherapy |
| C | Comparison | Patients with head and neck cancer |
| O | Outcome | Survival of dental implants |

implants, implantation site, time interval between radiotherapy and implant installation, follow-up period, implant characteristics, cancer characteristics, adjuvant therapy, bone crest stability, postoperative complications, prosthetic rehabilitation. Subsequently, a third author (DVC) reviewed the data. In cases of missing data, authors would be contacted by email twice.

## 2.6 Assessment of methodological quality

The quality of included systematic reviews was appraised by using the A Measurement Tool to Assess Systematic Reviews (AMSTAR 2) tool [34], and it was independently assessed by two reviewers (MCDP and LCA). Disagreements were resolved by a third reviewer (DVC). The methodological quality of systematic reviews was assessed using the, which consists of 16 items, where items 2, 4, 7, 9, 11, 13, and 15 are considered critical. Questions are answered with options: yes, partial yes, no, or meta-analysis not conducted. The overall confidence rating of the review's results is categorized as high (none or one non-critical weakness), moderate (more than one non-critical weakness), low (one critical flaw with or without non-critical weaknesses), or critically low (more than one critical flaw with or without non-critical weaknesses).

## 2.7 Results synthesis

A descriptive synthesis of the results of the included studies was conducted the percentage of implant survival was calculated by arithmetic mean.

# 3 Results

## 3.1 Study selection

The PRISMA flowchart depicting the study selection process is presented in Fig 1. The literature search strategy identified 1,811 articles. After removing 670 duplicates, the remaining 1,141 publications were evaluated by reading titles/abstracts. A total of five additional reports were identified through manual search. This resulted in 45 records selected for reading the full text. Thirty-four references were excluded because 11 did not investigate the control group - non-irradiated [35–45], eight were not systematic reviews [46–53], in 7 references the full text was unavailable [54–60], three did not focus on oral rehabilitation with implants in irradiated patients [61–63], two included nonspecific data for head and neck cancer [64,65]; two were editorials [66,67], and one did not investigate implant survival [68]. Finally, eleven systematic reviews were included for qualitative synthesis. A table of all studies identified in the literature search, including those that were excluded from the analyses with the reasons for exclusion is presented in S1 Text, S2 Table.

## 3.2 Main characteristics of included reviews

Tables 1 and 2 show the characteristics of the eligible systematic reviews. A total of 16,193 patients who received 73,674 dental implants were included. Of these, approximately 31,137 were installed in irradiated sites. All selected studies [15,24–26,69–74] assessed the survival of dental implants installed after the diagnosis of head and neck cancer. One systematic review [25] was an update of a previous study elaborated by the same research team [75]. The age range of participants varied from 6 to 94 years, with one study not reporting the sample age [74]. Among the included reviews, most evaluated data from retrospective studies. All included studies conducted meta-analysis.

The minimum radiation dose among the studies ranged from 10 Gy to 40 Gy and the maximum dose from 72 Gy to 145 Gy; two studies did not mention the radiation dose [70,74].

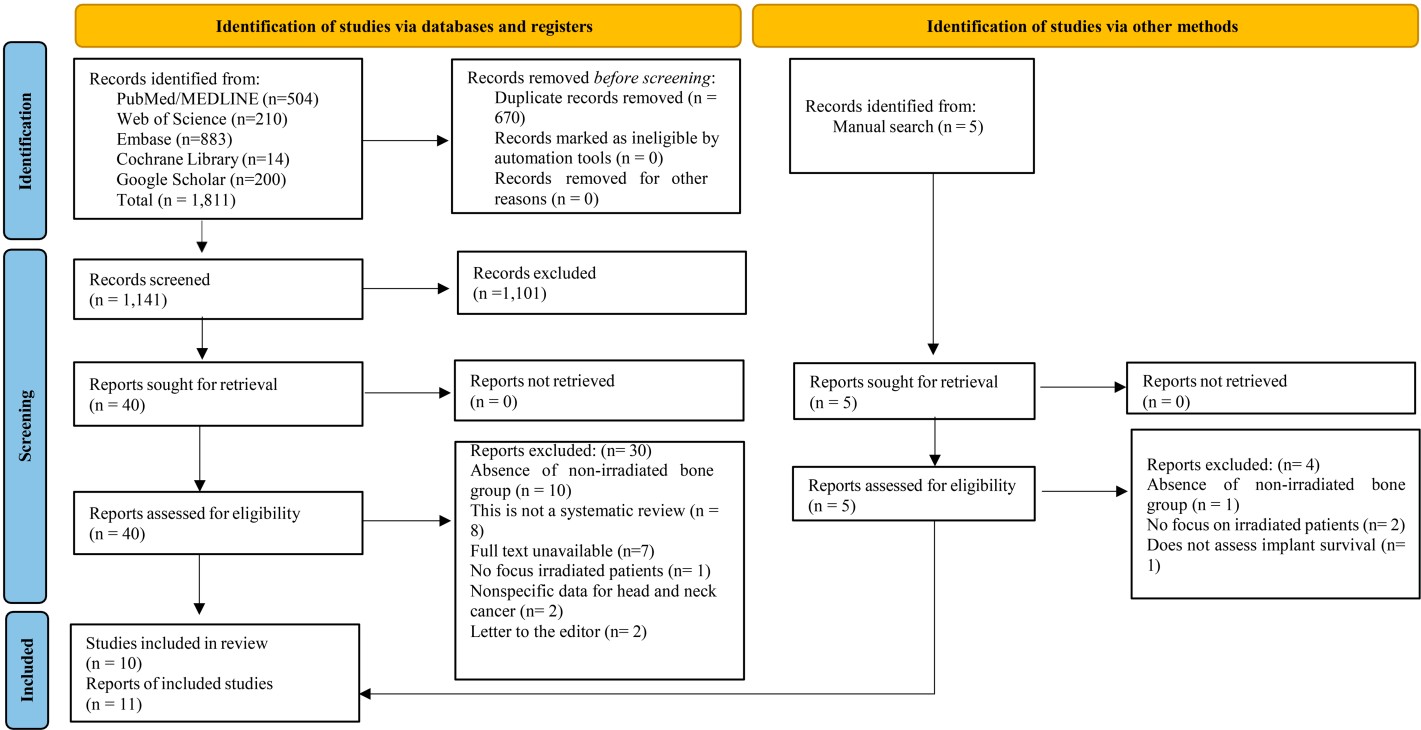

**Fig 1. PRISMA 2020 flow diagram for new systematic reviews which included searches of databases, registers, and other sources.**

Regarding the timing of implant installation, five articles described implants installed before and after radiotherapy [15,25,26,73,74], four reported implants installed after radiotherapy [69–72]; one study evaluated implant installation during panendoscopy or ablative surgery - before radiotherapy [24]. All selected studies evaluated the survival of implants installed in the maxilla and mandible. While seven studies investigated implants installed in native and grafted bone [15,24–26,69,72,73], two analyzed implants installed only in native bone [71,74], and one study did not mention [70].

Only one study investigated the time interval between radiotherapy and implantation [15]. The follow-up time among the studies varied from one month to 23 years, with one study not mentioning the follow-up period [70]. More details regarding the study design, number of implants installed, number of failed implants, and implant survival percentage extracted from the primary articles are described in S1 Text, S3 Table.

## 4 Synthesis of results

### 4.1 Survival percentage of dental implants in irradiated and non-irradiated patients

The survival rate of implants was higher among non-irradiated patients according to data from 11 meta-analyses. Conversely, three meta-analyses showed no significant difference between the irradiated and non-irradiated groups [24,26,75]. Data were reported in four time periods (1990-2021, 1990-2006, 2007-2013, and 2013-2021) and two follow-up periods (≥3 years of follow-up and ≥5 years of follow-up) [75].

**Table 2. General characteristics of the included systematics reviews (part 1).**

| Study (year) | N Participants/Age range (years) | Country | N Primary studies/N Study design | Presence of metaanalysis | Radiation dose range (Gy)/Time of implant placement | *N Failed/placed implants | Implantation site | Interval between RT and implant placement, months | Follow-up, mo |
|---|---|---|---|---|---|---|---|---|---|
| Camolesi et al. (2023) | 876/31,6% (>60) | Espanha | 19/PS:2 RS:17 | Yes | 17 S (≥ 50 Gy) 2 S (<50 Gy)/Post RT | Overall: 4,473 | 1 S (Mx) 7 S (Md) 8 S(Mx/md) NB/GB | 6-12 Pot RT 13-20 Posr RT >21 Post RT | 5 S (>5 years) |
| Shahi et al (2023) | 726/50,93 mean | Iran | 13/RS: 13 | Yes | NM/Post RT | 136/1,226(EG) 41/932(CG) | Mx/md | NM | NM |
| Kende et al. (2022) | 1,246/51.43 mean | India | 23/RS:21 PS:2 | Yes | 40-72/**Pre and post RT | 364/2,186 (EG) 179/1,685 (CG) | Mx/md NB/GB | NM | 52.5 mean |
| Schiegnitz et al. (2022) | 3,445/6-91 | Germany | 59/PS:12 RS:47 CSS:1 | Yes | 20-81,6/Pre and post RT | 373/1,825(EG) 278/2,077 (CG) | Mx/md NB/GB | NM | 3-120 |
| Shokouhi and Cerajewska (2022) | 441/28-88 | United Kingdom | 07/PS:4 RS:3 | Yes | 10-72/Post RT | 62/508(EG) 13/452 (CG) | Mx/md NB | NM | 12-168 |
| Gupta et al. (2021) | 1,097/13-89 | Italy | 15/RS:5 PS:10 | Yes | 20-72/Post RT | 323/2,203 (EG) 141/2,349 (CG) | Mx/md NB/GB | ***6-41 Post RT | 6-120 |
| In't Veld et al. (2021) | 1,148/17-91 | Netherlands | 10/PS:6 RS:4 | Yes | 30-72/Pre RT | 37/413 (EG) 5/372 (CG) | Md/mx NB/GB | Not performed | 12-174 |
| Smith Nobrega et al. (2016) | 2,220/58 mean | Brazil | 40/PS:18 RS:21 RCT:1 | Yes | 10-145/Pre and post RT | 629/3,773 (EG) 191/3,228 (CG) | Mx/md NB/GB | ***1-240 Post RT | 1-276 |
| Chrcanovic et al. (2016) | 4,431/6-94 | Sweden | 54/RS:44 CCT:10 | Yes | 21-120/Pre and post RT | 640/3,914 (EG) 684/14,514 (CG) | Mx/md NB/GB | <12 Post RT ≥12 Post RT | Up to 180 |
| Chambrone et al. (2013) | 563/NM | Brazil | 15/RS:11 PS:2 RCT:2 | Yes | NM/Pre and post RT | 211/1,689 (EG) 305/8,461 (CG) | Mx/md NB | NM | Up to 181 |

Abbreviations: CG: control group (nonirradiated patients); CCT: controlled clinical trial; CS:cross sectional study; EG: experimental group (irradiated patients); GB: grafted bone; mo: months; N: number; NB: native bone; Mx/md: maxilla and mandible; NM: not mentioned; PS: prospective study; RS: retrospective study; RCT: randomized controlled trial; RT: radiotherapy; S: studies. * Numbers reported in meta-analyses including data from the longest follow-up period. ** Implants placed during tumor ablative surgery prior to radiotherapy and implants placed after completion of radiotherapy. *** Variation in primary studies.

The effect estimate and precision of each meta-analysis are presented in Table 3. Further information on the number of failed/installed implants, number of primary articles included in the meta-analysis, and the percentage of survival is also shown in S4 Table (S1 Text).

Overall, 48,563 implants were evaluated, including 14,471 with 2,674 failures in the irradiated group and 34,092 with 1,825 failures in the non-irradiated group. The survival percentage was 81.52% and 94.64%, respectively.

## 4.2 Survival percentage of implants installed in irradiated maxilla versus irradiated mandible

Two meta-analyses [15,74] showed higher implant survival installed in irradiated mandible compared to irradiated maxilla (RR: 3.16; 95% CI: 1.76-5.68; p = 0.0001; I2 = 70%) (RR: 5.96; 95% CI: 2.71-13.12; p0.00001; I2 = 33%). A total of 3,378 implants were evaluated, including 897 with 188 failures in the irradiated maxilla group and 2,481 with 160 failures in the irradiated mandible group. The survival percentage was 79.04% and 93.55%, respectively.

In the study by Shokouhi and Cerajewska (2022), implants installed in the maxilla were significantly more likely to fail than those installed in the mandible (OR: 5.03; 95% CI: 1.07-23.58; p = 0.04; I2 = 87%).

**Table 3. General characteristics of the included systematics reviews (part 2).**

| Study (year) | Implant characteristics | Characteristics of cancer | Adjuvant therapy | Bone crest stability | Postoperative complications | Prosthetic rehabilitation |
|---|---|---|---|---|---|---|
| Camolesi et al. (2023) | NM | Head and neck cancer | 8 S (CT+ surgery) | NM | 7 S (PI) 5 S (ORN) | NM |
| Shahi et al. (2023) | NM | Oral cancer | NM | NM | NM | NM |
| Kende et al. (2022) | NM | Oral cancer | NM | NM | NM | NM |
| Schiegnitz et al. (2022) | NM | Head and neck cancer | CT HBO AT | NM | ORN, PI, Mucositis | NM |
| Shokouhi and Cerajewska (2022) | NM | Head and neck cancer | NM | NM | ORN | NM |
| Gupta et al. (2021) | Branemark, Frialit, TPS, SLA, ITI, IMZ, Xive, Astra tech, Steri-Oss, Camlog, Osteotite implants | NM | HBO AT | NM | NM | NM |
| In't Veld et al. (2021) | Osseospeed/Astra tech, Branemark machined surface or a Ti-Unite surface/Nobel Biocare, Branemark MK II/III 2-phase implants/Nobel Biocare, Frialit 2-phase implants/Nobel Biocare, ITI/Straumann, Neoss/Neoss implant | Head and neck cancer | HBO | NM | ORN, PI, LO, Tumor recurrence | Overdenture |
| Smith Nobrega et al. (2016) | Astra tech, Nobel biocare, Serf, Friatec, Lifecore biomedical, Straumann, Osseous-mozograu, Sterio Os, Camlog, Friadent, Interpore, Internatonal, Imtec, Dyna, Screw-vent | NM | HBO | NM | ORN, PI | NM |
| Chrcanovic et al. (2016) | Branemark, Bonefit, IMZ, Frialit-2 system, Astra, Bonefit, Steri-Oss, ITI, Ankylos, Dyna, Ciny, SLA, CAMLOG, Branemark MKII, MKIII, Endopore, MG Osseus-mozograu, TiUnite, Neoss | Head and neck cancer | CT AT MR HBO | MBL | Postoperative infection | Fixed overdenture |
| Chambrone et al. (2013) | Branemark, TPS, SLA, SLActive | Head cancer | HBO | NM | NM | NM |

Abbreviations: AT: antibiotic therapy, BOT; base of tongue, CT: chemorthertapy, DM: dorsal maxila, FOM: floor of mouth, HBO: hyperbaric oxygen therapy, LO: lack of osseointegration, LP: laryngopharynx, MBL: marginal bone loss, MR: mouth rinse, NM: not mentioned, NP: nasopharynx, OP: oropharynx, ORN: osteoradionecrosis, PI: periimplantitis, S: studies; SLA: sandblasted and acid etched, SLActive: modified sandblasted and acid etched, TPS: titanium plasma sprayed.

## 4.3 Survival of implants installed in irradiated grafted bone versus irradiated native bone

Two meta-analyses [25] – literature 1990-2021 – showed higher implant survival in irradiated native bone. (OR: 2.06; 95% CI: 1.44-2.94; p0.0001; I2 = 0% - follow-up ≥ 36 months) (OR: 2.26; 95% CI: 1.50-3.40; p0.0001; I2 = 12% - follow-up ≥ 60 months). Conversely, one meta-analysis [15] showed no significant difference between irradiated grafted bone and irradiated native bone (RR: 1.35; 95% CI: 0.93-1.94; p = 0.11; I2 = 48%). A total of 1,740 implants were evaluated, including 596 with 106 failures in the irradiated grafted bone group and 1,144 with 148 failures in the irradiated native bone group. The survival percentage was 82.21% and 87.06%, respectively.

A meta-analysis showed that there was a tendency for implants subjected to lower doses of radiotherapy to have a higher implant survival (RR: 1.59; 95% CI: 0.98-2.59; p = 0.06; I2 = 47%). However, no statistically significant difference was observed between the groups evaluated [15]. A total of 1,057 implants were evaluated, including 546 with 98 failures in the higher dose group and 511 with 51 failures in the lower dose group. The survival percentage for implants subjected to high doses was 82.05% and for lower doses it was 90.01%. A meta-analysis showed that no statistically significant difference was observed when implants

were installed before or after 12 months of radiotherapy [15], for 445 implants installed up to 12 months after radiotherapy, 79 failures occurred with a survival percentage of 84.26%, and in the group of implants installed after 12 months of radiotherapy application, 58 failures occurred in 505 implants with a survival percentage of 88.51% (RR: 1.37; 95% CI: 0.76-2.45; p = 0.29; I2 = 64%).

## 4.4 Methodological quality of systematic reviews

Overall, according to the evaluated domains using the AMSTAR 2 tool, the eligible reviews exhibited critically low quality (Table 4). The justifications for downgrading the studies are detailed in the S1 Text, S4 Table.

## 5 Discussion

The present umbrella review included eleven systematic reviews published between 2013 and 2023, assessing the survival of 14,471 dental implants installed before and/or after radio-therapy in head and neck cancer patients, compared to 34,092 implants in non-irradiated patients. Overall, the survival rate of implants installed in irradiated patients was considered inferior compared to those installed in non-irradiated patients. Implant survival in

**Table 4. Summary of meta-analyses on implant failures in irradiated versus nonirradiated patients.**

| Meta-analysis | N Failed/placed implants | Results | N Studies Survival rate (%) |
|---|---|---|---|
| Camolesi et al. (2023) 5 years of follow-up | NM | WP (EG): 93.13%; CI 95%: 87.20-99.6; $p < 0.001$; $I^2$ = 85% WP (CG): 98.52%; CI 95%: 97.56-99.48; $p < 0.001$; $I^2$ = 0% | S: 3 93.13(EG) 98.56(CG) |
| Shahi et al. (2023) follow-up not mentioned | 136/1,226(EG) 41/932(CG) | OR: 1.79; CI 95%: 1.35-2.23; $p < 0.01$; $I^2$ = 62, 89% | S: 9 81.35(EG) 87.45(CG) |
| Kende et al. (2022) mean 52.5 months of follow-up | 364/2,186(EG) 179/1,685(CG) | OR: 0.24; CI 95%: 0.14-0.40; $p < 0.01$; $I^2$ = 73% | S: 16 82.47(EG) 89.37(CG) |
| Schiegnitz et al. (2022) (20132021) ≥ 3 years of follow-up | NM | OR: 2.07; CI 95%: 1.54-2.97; $p < 0,00001$ | NM |
| Schiegnitz et al. (2022) (20132021) ≥ 5 years of follow-up | NM | OR: 1.8; CI 95%: 1.21-2.67; $p = 0.003$ | NM |
| Schiegnitz et al. (2022) (19902021) 3 years of follow-up | 500/3,269(EG) 339/3,348(CG) | OR: 2.06; CI 95%: 1.75-2.42; $p < 0.00001$; $I^2$ = 33% | S: 21 79.56(EG) 86.61(CG) |
| Schiegnitz et al. (2022) (1990 - 2021) 5 years of follow-up | 373/1,825(EG) 278/2,077(CG) | OR: 1.97; CI 95%: 1.63-2.37; $p < 0.00001$; $I^2$ = 47% | S: 12 84.70(EG) 89.87(CG) |
| Shokouhi and Cerajewska (2022) 1-3.8 years of followup | 62/508(EG) 13/452(CG) | OR: 4.77; CI 95%: 2.57-8.89; $p < 0.00001$; $I^2$ = 0% | S: 4 87.79(EG) 97.12(CG) |
| Gupta et al. (2021) 6-12 months of follow-up | 323/2,203(EG) 141/2,349(CG) | OR: 2.95; CI: 1.93-4.50; $p < 0.00001$; $I^2$ = 50% | S: 14 85.80(EG) 93.90(CG) |
| In't Veld et al. (2021) followup not mentioned | 37/473(EG) 5/372(CG) | RR: 5.02; CI 95%: 0.92-27.38; $p = 0.06$; $I^2$ = 56% | S: 3 92.17(EG) 98.65(CG) |
| Smith Nobrega et al. (2016) 16 years of folow-up | 629/3,773(EG) 191/3,328(CG) | RR: 2.63; CI 95%: 1.93-3.58; $p < 0.001$; $I^2$ = 56% | S: 26 84.30(EG) 94.26(CG) |
| Chrcanovic et al. (2016) follow-up not mentioned | 640/3,914(EG) 684/14,514(CG) | RR: 2.18; CI 95%: 1.71-2.79; $p < 0.00001$; $I^2$ = 52% | S: 37 83.64(CG) 95.29(EG) |
| Shiegnitz et al. (2014) (20072013) ≥ 5 years of follow-up | 19/263(EG) 13/241(CG) | OR: 1.44; CI 95%: 0.67-3.10; $p = 0, 35$; $I^2$ = 75% | S: 3 92.77(EG) 94.60(CG) |
| Shiegnitz et al. (2014) (19902006) ≥ 5 years of follow-up | 281/983(EG) 203/1,040(CG) | OR: 2.11; CI 95%: 1.69-2.65; $p < 0.00001$; $I^2$ = 72% | S: 4 71.41(EG) 80.48(CG) |
| Chambrone et al. (2013) follow-up not mentioned | 110/549(EG) 293/8,383(CG) | RR: 2.74; CI 95%: 1.86-4.05; $p < 0.00001$; $I^2$ = 0% | S: 7 83.30(EG) 93.81(CG) |

Abbreviations: CG: control group (nonirradiated patients); CI: confidence interval; EG: experimental group (irradiated patients); N: number; NM: not mentioned; OR: odds ratio; RR: risk ratio; S: studies; WP: weighted proportion

the irradiated maxilla was significantly lower compared to the irradiated mandible. Dental implants installed in irradiated grafted bone were considered more prone to failure compared to irradiated native bone. The methodological quality of the selected studies was considered critically low (Table 5).

Evidence from systematic reviews supports that implant survival is influenced by the effect of radiotherapy [64,65]. Zarzar et al., (2024) [23], in an umbrella review, demonstrated that the implant survival rate in irradiated patients was lower compared to non-irradiated patients, with follow-up ranging from 1 month to 23 years. The present umbrella review evaluated and summarized scientific evidence available in the literature, corroborating previous reviews. It is important to mention that the analyzed data had different follow-up periods, which could cause significant errors in the descriptive synthesis presented in this study.

The impact of radiation dose on the survival of dental implants installed in head and neck cancer patients depends on the specific dose at the implant site (>50 Gy) and the parotid gland (>30 Gy) [19]. An umbrella review reported that radiation doses lower than 50 Gy were associated with better survival rates [22]. This is consistent with the results observed in this umbrella review. It is important to consider that these data compare high doses with low doses, not detailing the implantation site with the target volume of radiation. Moreover, the systematic reviews included studies that varied the maximum doses from 72 Gy to 145 Gy, which may represent a significant risk of implant failure and osteoradionecrosis [76].

Most implants evaluated in this review were installed after the completion of radiotherapy. Implants installed after the end of radiotherapy were significantly more likely to fail than those installed in non-irradiated bone [71,72]. Whereas implants installed immediately during ablative surgery (pre-RT) showed no statistically significant difference between irradiated

**Table 5. Assessment of the methodological quality of systematic reviews using the AMSTAR 2 tool.**

| Systematic review | Q. 1 | Q. 2 | Q. 3 | Q. 4 | Q. 5 | Q. 6 | Q. 7 | Q. 8 | Q. 9 | Q. 10 | Q. 11 | Q. 12 | Q. 13 | Q. 14 | Q. 15 | Q. 16 | Overall classification |
|---|---|---|---|---|---|---|---|---|---|---|---|---|---|---|---|---|---|
| Camolesi et al. (2023) | Y | Y | Y | N | Y | Y | N | Y | Y | N | Y | Y | N | Y | Y | N | Critically low |
| Shahi et al. (2023) | Y | N | Y | N | N | N | N | N | Y | N | Y | N | N | Y | Y | Y | Critically low |
| Kende et al. (2022) | Y | Y | Y | N | Y | Y | N | N | Y | N | N | N | N | N | N | Y | Critically low |
| Schiegnitz et al. (2022) | N | PY | Y | N | N | N | N | Y | Y | N | Y | Y | Y | Y | Y | Y | Critically low |
| Shokouhi and Cerajewska (2022) | Y | N | Y | PY | N | N | N | Y | Y | N | Y | Y | Y | Y | N | Y | Critically low |
| Gupta et al. (2021) | Y | N | Y | N | Y | Y | N | Y | Y | N | Y | Y | N | Y | N | Y | Critically low |
| In't Veld et al. (2021) | Y | N | N | N | Y | N | N | Y | N | N | Y | Y | Y | Y | N | Y | Critically low |
| Smith Nobrega et al. (2016) | Y | N | Y | N | Y | Y | N | Y | N | N | Y | Y | Y | Y | Y | N | Critically low |
| Chrcanovic et al. (2016) | Y | N | Y | PY | Y | N | N | Y | N | N | Y | Y | Y | Y | Y | N | Critically low |
| Shiegnitz et al. (2014) | N | N | Y | N | Y | Y | N | Y | N | N | Y | N | Y | Y | Y | N | Critically low |
| Chambrone et al. (2013) | Y | N | Y | PY | Y | Y | Y | Y | Y | N | Y | Y | Y | Y | N | Y | Critically low |

Abbreviations: Y: yes, N: not, PY: Partially yes Q.1. Did the research questions and inclusion criteria for the review include the components of PICO? Q.2. Did the report of the review contain an explicit statement that the review methods were established prior to the conduct of the review and did the report justify any significant deviations from the protocol? Q. 3. Did the review authors explain their selection of the study designs for inclusion in the review? Q. 4. Did the review authors use a comprehensive literature search strategy? Q. 5. Did the review authors perform study selection in duplicate? Q. 6. Did the review authors perform data extraction in duplicate? Q. 7. Did the review authors provide a list of excluded studies and justify the exclusions? Q. 8. Did the review authors describe the included studies in adequate detail? Q. 9. Did the review authors use a satisfactory technique for assessing the risk of bias (RoB) in individual studies that were included in the review? Q. 10. Did the review authors report on the sources of funding for the studies included in the review? Q. 11. If meta-analysis was performed, did the review authors use appropriate methods for statistical combination of results? Q. 12. If meta-analysis was performed, did the review authors assess the potential impact of RoB in individual studies on the results of the meta-analysis or Other evidence synthesis? Q. 13. Did the review authors account for RoB in primary studies when interpreting/discussing the results of the review? Q. 14. Did the review authors provide a satisfactory explanation for, and discussion of, any heterogeneity observed in the results of the review? Q. 15. If they performed quantitative synthesis did the review authors carry out an adequate investigation of publication bias (small study bias) and discuss its likely impact on the results of the review? Q.16. Did the review authors report any potential sources of conflict of interest, including any funding they received for conducting the review?

and non-irradiated groups [24]. There is a trend to install the implant during tumor removal, significantly reducing prosthetic rehabilitation time and, on the other hand, increasing complications that may hinder implantation success, such as obtaining a well-positioned implant [5,77,78].

The time interval between radiotherapy and implant surgery may influence implant survival. A meta-analysis showed no statistically significant difference when comparing implant failures installed before and after 12 months of radiotherapy [15]. Claudy et al. (2013) [41], in a meta-analysis, evaluated the time interval between 6 and 12 months after radiotherapy versus 12 months after radiotherapy, and an increased risk of 34% was observed for implants installed between 6 and 12 months post-radiotherapy. The literature recommends implant installation at least six months after the completion of radiotherapy, the expected period for the end of the acute process in bone and soft tissues induced by radiation [21].

The umbrella reviews, Zarzar et al. (2024) and Marques et al. (2023), using the AMSTAR 2 tool, included five systematic reviews in common with this study [15,24,73–75]. Zarzar et al. (2024) classified the systematic reviews of Shiegnitz et al. (2014), Smith Nobrega et al. (2016), and In't Veld et al. (2021) as critically low, which is in line with the results observed in this study. In contrast, the systematic review by Chambrone et al. (2013) was considered high quality and Chrcanovic et al. (2016) as low quality. This difference could be explained by the complexity associated with judging the issues. Marques et al. (2023) analyses combined responses to the 16 individual items and performed an overall score to assess the risk of bias of the studies, making the classification criteria different from the assessments of this study.

The present study investigated irradiated head and neck cancer patients, recommending caution in interpreting the results, despite the high survival rate demonstrated for dental implants. The evaluation of methodological quality, through the AMSTAR 2 tool, showed that although the included studies reported the use of the PRISMA guideline, only two systematic reviews reported prior registration of the research protocol [26,69]. The PRISMA guideline itself advises indicating where the review protocol can be accessed to allow readers to assess whether any deviations from the protocol compared to what was done may have introduced bias.

Some limitations should be considered when interpreting the results of this umbrella review. The search strategy identified only eleven systematic reviews, all classified with critically low methodological quality. Considering the retrospective nature of most primary studies included in the selected systematic reviews, it was observed that the reported data were notably heterogeneous, especially for questions related to implant characteristics, radiation, and follow-up period. The effect measure used in conducting the selected systematic reviews was the number of implants that failed and the number of implants installed, not including details about the patient or radiation to answer the research question. Additionally, implant surface properties are confounding factors to be considered. In this sense, some studies [15,24,72–74] sought to evaluate the geometry and surface treatment of the implant. However, it was not possible to reach a consensus on the risk of implant failures based on their characteristics because most individual studies did not report these data.

The main finding of this study was that, for patients after the diagnosis of head and neck cancer, the survival of implants that were irradiated before and/or after implantation was lower than that of implants placed in non-irradiated patients. The level of evidence in this comprehensive review was impacted by several negative factors and showed a lack of prospective controlled clinical trials. At present, the findings in these studies are not reliable for clinical decision-making because there are no standardized classification systems for oral rehabilitation of irradiated head and neck cancer patients.

We recommend that future randomized clinical trials be conducted on this topic. On the other hand, we consider the difficulty involved in conducting controlled and randomized experiments in this population. Therefore, new prospective observational studies are needed, combining standardized clinical approaches that can generate more reliable data, for example: individual data on the characteristics of the implants and the radiation dose applied. The literature is poorly explored regarding the influence of radiotherapy on implants placed before the diagnosis of head and neck cancer; future research is also indicated in this regard.

## 6 Conclusion

Despite weak evidence on dental implant failure in irradiated patients, it is important to evaluate the survival of implants installed before and/or after radiotherapy in head and neck cancer patients, which may improve the quality of life for these patients. Radiotherapy was associated with a higher percentage of dental implant losses, especially for implants installed in the maxilla and grafted bone. Dental implants can be placed in patients who have undergone radiotherapy for head and neck cancer.

The clinical relevance of this research is related to the need for each case to be carefully planned by a multidisciplinary team in order to consider whether the additional risks, costs and increased rehabilitation time are worthwhile. In addition, it is necessary to emphasize the importance of achieving satisfactory oral hygiene, adequate dentures and keeping patients in frequent follow-up appointments.

There are many biases in the selected systematic reviews, and the strength of evidence for analysis is weak. Therefore, the results of this umbrella review are not reliable to provide security in clinical decision-making. New systematic reviews with high methodological quality are necessary to increase the power of evidence in the success of implant installation in irradiated patients with head and neck cancer.

## Supporting information

**S1 Text.S1–S4 Tables.**
(PDF)

## Author contributions

**Conceptualization:** Caio Fernando Teixeira Portela, Maria Cândida Dourado Pacheco, Danilo Viegas da Costa.

**Formal analysis:** Maria Cândida Dourado Pacheco, Alexandre Henrique dos Reis Prado.

**Investigation:** Maria Cândida Dourado Pacheco, Danilo Viegas da Costa.

**Methodology:** Maria Cândida Dourado Pacheco, Danilo Viegas da Costa.

**Project administration:** Maria Cândida Dourado Pacheco.

**Resources:** Maria Cândida Dourado Pacheco.

**Software:** Caio Fernando Teixeira Portela.

**Supervision:** Maria Cândida Dourado Pacheco, Tarcísio Passos Ribeiro Campos, Arno Heeren de Oliveira.

**Validation:** Maria Cândida Dourado Pacheco, Alexandre Henrique dos Reis Prado, Lara Cancella de Arantes.

**Visualization:** Caio Fernando Teixeira Portela, Maria Cândida Dourado Pacheco.

**Writing – original draft:** Caio Fernando Teixeira Portela.

**Writing – review & editing:** Caio Fernando Teixeira Portela, Maria Cândida Dourado Pacheco.

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
