## [Decision Letter · Decision Letter 0]

20 Nov 2024

PONE-D-24-47771Survival of dental implants in irradiated head and neck cancer patients compared to non-irradiated patients: An umbrella reviewPLOS ONE

Dear Dr. Portela,

Thank you for submitting your manuscript to PLOS ONE. After careful consideration, we feel that it has merit but does not fully meet PLOS ONE’s publication criteria as it currently stands. Therefore, we invite you to submit a revised version of the manuscript that addresses the points raised during the review process.

We look forward to receiving your revised manuscript.

Kind regards,

Chung-Ta Chang

Academic Editor

PLOS ONE

Journal Requirements:

3. We note that your Data Availability Statement is currently as follows: “All relevant data are within the manuscript and in Supporting Information files.”

6. As required by our policy on Data Availability, please ensure your manuscript or supplementary information includes the following:

Reviewers' comments:

Reviewer's Responses to Questions

**Comments to the Author**

1. Is the manuscript technically sound, and do the data support the conclusions?

Reviewer #1: Yes

Reviewer #2: Yes

2. Has the statistical analysis been performed appropriately and rigorously? 

Reviewer #1: N/A

Reviewer #2: No

3. Have the authors made all data underlying the findings in their manuscript fully available?

Reviewer #1: Yes

Reviewer #2: Yes

4. Is the manuscript presented in an intelligible fashion and written in standard English?

Reviewer #1: Yes

Reviewer #2: Yes

5. Review Comments to the Author

Reviewer #1: Dear authors,

I evaluated the article titled "Survival of dental implants in irradiated head and neck cancer patients compared to non-irradiated patients: An umbrella review”.

The goal of this “umbrella review aimed to analyse the available relevant systematic reviews, comparing irradiated versus non-irradiated patients, in an attempt to determine whether there is an influence of head and neck radiotherapy on dental implant survival in patients with head and neck cancer."

INTRO

- “(Jung et al., 2012) [1].” Please update this ref. (below)

(new ref) Zirconia Implants and Marginal Bone Loss: A Systematic Review and Meta-Analysis of Clinical Studies. International Journal of Oral & Maxillofacial Implants, v. 35, p. 707-720, 2020.

(new ref) Histological osseointegration level comparing titanium and zirconia dental implants: Meta-analysis of pre-clinical studies. Int J Oral Maxillofac Implants. 2023; 38(4):667-680. doi: 10.11607/jomi.10142

- “However, the risks associated with implant failures have been extensively investigated over the years (Papi et al., 2019; Albrega et al., 2020; Pieralli et al., 2021; Li et al., 2022) [2, 3, 4, 5]” - the author used 4 references reporting only failures related to patients with cancer and irradiation; this fact is not transmitting a correct message. Change/include articles below to enrich it.

(new ref.) Marginal Bone Level and Clinical Parameter Analysis Comparing External Hexagon and Morse Taper Implants: A Systematic Review and Meta-Analysis. Diagnostics, 2023, 13, 1587. https://doi.org/10.3390/diagnostics13091587

(new ref) Bisphosphonates and Their Influence on the Implant Failure: A Systematic Review. Appl. Sci. 2023, 13, 3496. https://doi.org/10.3390/app13063496

(new ref) Marginal Bone Level and Biomechanical Behavior of Titanium-Indexed Abutment Base of Conical Connection Used for Single Ceramic Crowns on Morse-Taper Implant: A Clinical Retrospective Study. J. Funct. Biomater 2023, 14, 128. https://doi.org/10.3390/jfb14030128

(new ref) Clinical performance comparing titanium and titanium-zirconium or zirconia dental implants: A systematic review of randomized controlled trials. Dentistry Journal 2022, 10, 83. doi: 10.3390/dj10050083

M&M

- the exclusion criteria is incomplete.

Results, Discussion, and Conclusion: This sections were very well presented, with relevant information included in the tables; the pictures presented also were relevant.

Reviewer #2: The authors should have carried out a meta-meta-analysis, as it is not correct to make an arithmetic mean of survival cases of Survival percentage of dental implants in irradiated and non-irradiated patients. A meta-meta-analysis is a meta-analysis of multiple meta-analyses.

6. PLOS authors have the option to publish the peer review history of their article (what does this mean?). If published, this will include your full peer review and any attached files.

Reviewer #1: No

Reviewer #2: No

---

## [Decision Letter · Decision Letter 1]

2 Jan 2025

PONE-D-24-47771R1Survival of dental implants in irradiated head and neck cancer patients compared to non-irradiated patients: An umbrella reviewPLOS ONE

Dear Dr. Portela,

Thank you for submitting your manuscript to PLOS ONE. After careful consideration, we feel that it has merit but does not fully meet PLOS ONE’s publication criteria as it currently stands. Therefore, we invite you to submit a revised version of the manuscript that addresses the points raised during the review process.

We look forward to receiving your revised manuscript.

Kind regards,

Chung-Ta Chang

Academic Editor

PLOS ONE

Reviewers' comments:

Reviewer's Responses to Questions

**Comments to the Author**

1. If the authors have adequately addressed your comments raised in a previous round of review and you feel that this manuscript is now acceptable for publication, you may indicate that here to bypass the “Comments to the Author” section, enter your conflict of interest statement in the “Confidential to Editor” section, and submit your "Accept" recommendation.

Reviewer #3: (No Response)

Reviewer #4: All comments have been addressed

Reviewer #5: (No Response)

2. Is the manuscript technically sound, and do the data support the conclusions?

Reviewer #3: Yes

Reviewer #4: Yes

Reviewer #5: Yes

3. Has the statistical analysis been performed appropriately and rigorously? 

Reviewer #3: Yes

Reviewer #4: Yes

Reviewer #5: Yes

4. Have the authors made all data underlying the findings in their manuscript fully available?

Reviewer #3: Yes

Reviewer #4: Yes

Reviewer #5: Yes

5. Is the manuscript presented in an intelligible fashion and written in standard English?

Reviewer #3: Yes

Reviewer #4: Yes

Reviewer #5: Yes

6. Review Comments to the Author

Reviewer #3: The manuscript addresses an important topic and follows a rigorous methodology. However, the introduction and discussion need improved readability, with a clearer focus on clinical implications. The inclusion of grey literature requires justification, and tables should be refined for clarity and alignment. Balance the critique of low-quality studies with practical relevance and actionable recommendations for future research. Ensure consistent referencing and formatting throughout.

Click on the highlighted portions in the manuscript to see detailed comments in the notes.

Reviewer #4: Comments: The manuscript can be accepted after few minor grammatical corrections as it technically provides sound piece of scientific research with data that supports the conclusions.

Reviewer #5: Thank you for your efforts in reviewing the evidence regarding implant survival in head and neck cancer patients, with and without radiation therapy.

I have the following formatting suggestions:

- Line 45: correct typo in "head and neck"

- Line 104-119: Revise the run-on sentence discussing excluded studies to enhance readability, such as a bulleted list.

- Line 160: Revise incomplete introduction sentence. It seems like this was supposed to be a paragraph heading, possible overlooked formatting.

Suggestions for content:

- Discussion Section: You discuss the findings of the previous literature regarding time interval's effect on implant survival. Please also provide your thoughts on this in light of your results suggesting no influence on implant survival.

7. PLOS authors have the option to publish the peer review history of their article (what does this mean?). If published, this will include your full peer review and any attached files.

Reviewer #3: **Yes: **AMITHA BASHEER N

Reviewer #4: **Yes: **DR AJAY SINGH RAO

Reviewer #5: **Yes: **Emma E. Kaz Frick

---

## [Author Response · Author response to Decision Letter 2]

18 Apr 2025

Reviewer #3: Amitha Basheer N

Thank you very much for your help and correction of the article. We hope we were able to correct.

Reviewer #3: The manuscript addresses an important topic and follows a rigorous methodology. However, the introduction and discussion need improved readability, with a clearer focus on clinical implications. The inclusion of grey literature requires justification, and tables should be refined for clarity and alignment. Balance the critique of low-quality studies with practical relevance and actionable recommendations for future research. Ensure consistent referencing and formatting throughout.

Click on the highlighted portions in the manuscript to see detailed comments in the notes.

Comments: The manuscript can be accepted after few minor grammatical corrections.

The suggestion is implemented and structured in the other corrections requested by other reviewers as well. The necessary changes and corrections requested have been sent in the article, we hope we have been able to live up to the occasion.

Authors' concluding remarks: Endnote was used to remove all duplicate references at once. For item 2.7 Results Synthesis, an excerpt from the text was removed due to the use of the arithmetic mean of the raw data to find the percentage.

Reviewer 5 Emma E. Kaz Frick

Reviewer #5: Thank you for your efforts in reviewing the evidence regarding implant survival in

head and neck cancer patients, with and without radiation therapy.

I have the following formatting suggestions:

- Line 45: correct typo in "head and neck"

Line 45, typo: "head nd neck cancer"

Done the modification.

- Line 104-119: Revise the run-on sentence discussing excluded studies to enhance

readability, such as a bulleted list.

“Thirty-four references were excluded because’’

Done the modification.

“In total, thirty-four references were excluded 11 did not investigate the control group - nonirradiated according to table 3.1, [25, 26, 27, 28, 29, 30, 31, 32, 33, 34, 35].”

- Line 160: Revise incomplete introduction sentence. It seems like this was supposed to be a

paragraph heading, possible overlooked formatting.

” Survival percentage of dental implants in irradiated and non-irradiated patients.”

Is the new section, thanks for correction. This excerpt is a subtitle and has been structured.

Thank you for your help in correcting it.

Suggestions for content:

- Discussion Section: You discuss the findings of the previous literature regarding time

interval's effect on implant survival. Please also provide your thoughts on this in light of your

results suggesting no influence on implant survival.

The suggestion is implemented and structured in the other corrections requested by other

reviewers as well.

Authors' concluding remarks: Endnote was used to remove all duplicate references at once.

For item 2.7 Results Synthesis, an excerpt from the text was removed due to the use of the

arithmetic mean of the raw data to find the percentage.

Reviewer #4: Dr. Ajay Singh Rao

Thank you very much for your help and correction of the article. We hope we were able to

correct.

Reviewer #4: Comments: The manuscript can be accepted after few minor grammatical

corrections as it technically provides sound piece of scientific research with data that supports

the conclusions.

Comments: The manuscript can be accepted after few minor grammatical corrections.

The suggestion is implemented and structured in the other corrections requested by other

reviewers as well.

Authors' concluding remarks: Endnote was used to remove all duplicate references at once.

For item 2.7 Results Synthesis, an excerpt from the text was removed due to the use of the

arithmetic mean of the raw data to find the percentage.

---

## [Editor Report · Decision Letter 2]

24 Apr 2025

Survival of dental implants in irradiated head and neck cancer patients compared to non-irradiated patients: An umbrella review

PONE-D-24-47771R2

Dear Dr. Portela,

We’re pleased to inform you that your manuscript has been judged scientifically suitable for publication and will be formally accepted for publication once it meets all outstanding technical requirements.

Kind regards,

Chung-Ta Chang

Academic Editor

PLOS ONE
---

## [Editor Report · Acceptance letter]

PONE-D-24-47771R2

PLOS ONE

Dear Dr. Portela,

I'm pleased to inform you that your manuscript has been deemed suitable for publication in PLOS ONE. Congratulations! Your manuscript is now being handed over to our production team.

Kind regards,

on behalf of

Dr. Chung-Ta Chang

Academic Editor

PLOS ONE